# Phytochemical Characterisation and Antioxidant and Anti-Inflammatory Potential of *Muscari neglectum* (Asparagaceae) Bulbs

**DOI:** 10.3390/molecules30224351

**Published:** 2025-11-10

**Authors:** María del Carmen Villegas-Aguilar, Antonio Segura-Carretero, Víctor N. Suárez-Santiago

**Affiliations:** 1Department of Analytical Chemistry, University of Granada, 18071 Granada, Spain; marivillegas@ugr.es; 2Department of Botany, University of Granada, 18071 Granada, Spain

**Keywords:** *Muscari neglectum*, phytochemical characterisation, flavonoids, antioxidant activity, anti-inflammatory activity

## Abstract

*Muscari neglectum* is a Mediterranean geophyte with a long tradition of ethnomedicinal use, yet the phytochemistry of its bulbs remains underexplored compared with aerial parts. This study aimed to characterise the metabolite profile of *M. neglectum* bulbs and to assess their antioxidant and radical scavenging potential, and anti-inflammatory potential. Bulb extracts were obtained by hydroethanolic extraction and analysed through HPLC-ESI-qTOF-MS, leading to the annotation of 72 compounds spanning diverse chemical families, including flavonoids, hydroxycinnamic acids, terpenoids, fatty acids, and triterpenoid saponins. Flavonoids constituted the most abundant group, with homoisoflavanones representing a characteristic class of metabolites in the *Muscari* genus and reflecting its distinctive secondary metabolism. Quantitative analyses revealed a high total phenolic content (65.5 mg GAE/g DE) and total flavonoid content (14.3 mg Epi/g DE). Antioxidant assays demonstrated measurable reducing power (FRAP: 0.26 mmol Fe^2+^/g DE; TEAC: 0.45 mmol TE/g DE), while radical scavenging assays indicated activity against superoxide anion (IC_50_ = 848 mg/L) and hypochlorous acid (IC_50_ = 9.2 mg/L). Additionally, the extract inhibited xanthine oxidase (IC_50_ = 20.6 mg/L). Furthermore, the extract exhibited significant anti-inflammatory activity, effectively scavenging nitric oxide radicals (IC_50_ = 78 ± 3 mg/L) and inhibiting lipoxygenase (IC_50_ = 66 ± 2 mg/L), suggesting that phenolic compounds and triterpenoid saponins contribute to the modulation of oxidative and enzymatic inflammatory pathways. These findings highlight *M. neglectum* bulbs as a rich source of structurally diverse bioactive compounds with antioxidant and anti-inflammatory capacity. The results provide a chemical basis for their traditional use and reinforce the value of bulb-specific studies within the Asparagaceae family.

## 1. Introduction

The genus *Muscari* Mill. (family Asparagaceae, subfamily Scilloideae), the grape hyacinths, comprises perennial bulbous herbs widely distributed across Europe, the Mediterranean basin, and parts of Western and Central Asia. Species of this genus are characterised by underground storage organs (bulbs), basal linear leaves, and small, spherical or urn-shaped flowers arranged in racemose inflorescences, commonly blue, violet, or white [1,2]. Beyond their ornamental value, *Muscari* species are recognised for their content of bioactive secondary metabolites, including flavonoids, homoisoflavanones, and saponins, which contribute to their ecological roles and potential pharmacological properties [3].

Plants of the genus *Muscari* have a longstanding history of use in Mediterranean and Balkan traditional medicine, where various species have been employed for their diuretic, anti-inflammatory, and digestive properties [4]. For instance, *Muscari neglectum* Guss. ex Ten. & Sangiovanni (Figure 1) and related species have been used in Turkish ethnomedicine for centuries to make decoctions that treat kidney diseases, ease gastrointestinal discomfort and aid detoxification [4]. Furthermore, *Muscari comosum* (L.) Mill. has a long history of use in the Mediterranean region, where its bulbs have been employed in folk medicine as remedies for toothache, as well as for their anti-inflammatory, diuretic, and aphrodisiac properties [5].

Although much of the ethnobotanical knowledge regarding *Muscari* species remains rooted in oral tradition, recent phytochemical studies have confirmed the presence of significant levels of phenolic compounds and flavonoids, supported their historical medicinal uses, and suggested promising antioxidant and anti-inflammatory potential. For instance, Evidence demonstrates the presence of flavonoids and homoisoflavonoids in *M. comosum* [6]. Similarly, extracts of *M. neglectum* display significant phenolic and flavonoid contents, which have been associated with antioxidant potential [7].

Despite the phytochemical characterisation available for *M. comosum* bulbs [6], detailed studies focusing specifically on the bulbs of *M. neglectum* remain scarce. Most previous research on *M. neglectum* has analysed flowers, aerial parts, or whole-plant extracts, potentially overlooking tissue-specific metabolite accumulation [8,9]. Bulbs, as storage organs, are known to concentrate distinct classes of secondary metabolites, including flavonoids and saponins, which may not be represented in aerial tissues [6]. Therefore, a targeted characterisation of bulb phytochemistry is essential to fully understand the bioactive potential of this species.

In addition to their antioxidant properties, *Muscari* species are also reported to exhibit anti-inflammatory effects, which may be mediated through the modulation of reactive nitrogen species and inflammatory enzymes [6]. Nitric oxide (NO) plays a dual role in inflammation, acting both as a signaling molecule and as a mediator of oxidative stress [10], while the enzyme lipoxygenase (LOX) catalyzes the oxidation of polyunsaturated fatty acids to pro-inflammatory lipid mediators [11]. Therefore, evaluating the NO radical scavenging capacity and the inhibition of LOX activity provides valuable insights into the potential anti-inflammatory mechanisms of *M. neglectum* bulb extracts.

In this context, the present study aims to provide a comprehensive characterisation of the phytochemical composition of *M. neglectum* bulbs and to evaluate their antioxidant, radical scavenging, and anti-inflammatory activities, thereby filling an important gap in the knowledge of this traditionally used medicinal plant.

## 2. Results

### 2.1. Preliminary Extraction Optimization

To optimize the extraction of phenolic compounds from *M. neglectum* bulbs, preliminary extractions were performed using different ethanol–water ratios (100:0, 80:20, 50:50, 20:80, 0:100, *v*/*v*) under the same extraction conditions. The total phenolic content (TPC) of the resulting extracts was determined using the Folin–Ciocalteu assay (Table 1). Absolute ethanol (100:0) yielded the highest TPC (96.3 ± 0.8 mg GAE/g DE). However, the 80:20 ethanol–water mixture was selected for all subsequent experiments, as it provides a substantial fraction of total phenolics (~70% of the 100:0 value) while favouring the extraction of more polar phenolic compounds, which are often biologically relevant. Moreover, hydroalcoholic extracts exhibit better solubility in aqueous assay media and are compatible with HPLC sample preparation, facilitating reproducible quantification and bioactivity testing. Practical considerations (safety, cost, and ease of solvent removal) further support the choice of 80:20 as the optimal extraction solvent. This selected 80:20 extract was subsequently employed for all further analyses, including antioxidant and anti-inflammatory assays, as well as for the comprehensive phytochemical characterization of *M. neglectum* bulbs using HPLC–ESI-qTOF-MS.

### 2.2. Phytochemical Characterization

The HPLC-ESI-qTOF-MS analysis of *M. neglectum* bulb extract revealed 72 molecular features after data processing. Most compounds could be tentatively annotated based on accurate mass, molecular formula, MS/MS fragmentation patterns, and database comparison, while a subset remained unidentified (Level 4). The base peak chromatogram (Appendix A) illustrates the chemical complexity of the extract, and Table 2 summarizes the identified and tentatively annotated compounds along with their retention times, proposed structures, and chemical families.

The annotated compounds belonged to a wide range of phytochemical families, including vitamins (e.g., pantothenic acid), amino acids (asparagine), organic acids (malic acid isomers, 3-hydroxysuberic acid), hydroxycinnamic acids (caffeic acid, dihydroferulic acid), coumarins (fraxin), flavonoids (homoisoflavanones, flavanones, flavonols, highly methoxylated flavones, glycosylated flavonoids), lignans (sesaminol 2-*O*-triglucoside), iridoid glycosides (torososide B, jasminoside S), fatty acids and oxylipin derivatives (linoleic acid, eicosapentaenoic acid, 9,12,13-trihydroxyoctadec-10-enoic acid), terpenoids (monoterpene derivatives), and triterpenoid saponins (ginsenosides Rh2 and Rh4, officinin A isomers, cucurbitacin P).

Officinin A isomers and ginsenoside Rh4 were annotated as MSI Level 2 compounds (HR-MS/MS matching), while cucurbitacin P and ginsenoside Rh2 were assigned as MSI Level 3 (accurate-mass-based). These metabolites have not been previously reported in *Muscari* and will be confirmed with authentic standards in future work.

Flavonoids were the most abundant class, including methoxylated and glycosylated derivatives such as dimethylquercetin, luteolin derivatives, hesperetin isomers, kaempferol 3-(2G-apiosylrobinobioside), and highly methoxylated flavones. Homoisoflavanones (muscomin and muscomosin isomers, 3′-hydroxy-3,9-dihydroeucomin) were also prevalent, reflecting the characteristic secondary metabolism of *M. neglectum* bulbs within the Asparagaceae family. Besides flavonoids, other bioactive metabolites were characterised, including hydroxycinnamic acids (caffeic acid, dihydroferulic acid), organic acids (malic acid isomers, 3-hydroxysuberic acid), coumarins (fraxin), and phenolic ketones (gingerol). Several fatty acids and derivatives, such as linoleic acid, eicosapentaenoic acid, and oxylipins (9,12,13-trihydroxyoctadec-10-enoic acid), were also identified. Iridoid glycosides (e.g., torososide B and jasminoside S) and triterpenoid saponins (e.g., ginsenosides Rh2 and Rh4, officinin A isomers, and cucurbitacin P) were annotated, further highlighting the chemical diversity of the extract.

A total of 18 signals remained unidentified, representing a subset of the detected features and emphasizing the complexity of the phytochemical composition of *M. neglectum* bulbs. Notably, several oxylipin derivatives were detected here for the first time in *M. neglectum* bulbs. The chemical diversity observed in the bulb extract prompted further evaluation of its antioxidant and radical scavenging properties, which are described in the following section.

### 2.3. Evaluation of Antioxidant and Antiradical Activities

TPC and total flavonoid content (TFC) of *M. neglectum* bulb extract were determined at 65.5 ± 0.3 mg GAE/g DE and 14.3 ± 0.3 mg Epi/g DE, respectively, with TPC approximately 4.5 times higher than TFC, indicating that non-flavonoid phenolics constitute a significant portion of the extract (Table 3). Antioxidant capacity assessed by FRAP and TEAC assays reached 0.26 ± 0.02 mmol Fe^2+^/g DE and 0.45 ± 0.02 mmol TE/g DE, respectively, reflecting measurable reducing and radical scavenging activity. Radical scavenging assays showed IC_50_ values of 848 ± 20 mg/L for superoxide anion (·O_2_^−^) and 9.2 ± 0.9 mg/L for hypochlorous acid (HOCl), while gallic acid and epicatechin, used as positive controls, displayed IC_50_ values of 50 ± 3 mg/L and 60 ± 1 mg/L for ·O_2_^−^, and 3.8 ± 0.3 mg/L and 0.18 ± 0.01 mg/L for HOCl, respectively (Table 4). The bulb extract also inhibited xanthine oxidase activity with an IC_50_ of 20.6 ± 0.4 mg/L, compared with 9 ± 1 mg/L for epicatechin (Table 5). Overall, these results provide a comprehensive quantitative description of the phenolic and flavonoid contents, as well as the antioxidant, antiradical, and enzymatic activities of the *M. neglectum* bulb extract, measured through multiple complementary assays.

### 2.4. Evaluation of Anti-Inflammatory Activity

The anti-inflammatory potential of the *M. neglectum* bulb extract was evaluated through its ability to scavenge ·NO radicals and to inhibit the enzyme LOX.

As shown in Table 6, the extract exhibited a moderate ·NO scavenging activity, with an IC_50_ value of 78 ± 3 mg/L, when compared to the reference antioxidants gallic acid (14.2 ± 0.8 mg/L) and epicatechin (8.7 ± 0.2 mg/L). Although the activity of the bulb extract was lower than that of the pure phenolic standards, these results demonstrate a relevant ability to neutralize reactive nitrogen species, suggesting that phenolic and saponin constituents may contribute synergistically to this effect.

Regarding LOX inhibition (Table 7), the *M. neglectum* bulb extract showed an IC_50_ value of 66 ± 2 mg/L, indicating a significant inhibitory effect on the lipoxygenase pathway. In comparison, the positive control nordihydroguaiaretic acid (NDGA) exhibited an IC_50_ of 3.63 ± 0.02 mg/L. Although the extract’s potency was lower than that of the standard inhibitor, it still indicates a meaningful capacity to modulate enzyme-mediated lipid peroxidation and inflammation.

## 3. Discussion

Previous phytochemical investigations on *M. neglectum* have been limited in scope. Early reports described the isolation of homoisoflavanones [15], while other studies analyzed the volatile fractions of bulbs and aerial parts by GC–MS, mainly identifying fatty acids and terpenoid constituents [8]. More recently, LC–MS/MS analyses of ethanolic extracts from whole plants or aerial tissues collected in Turkey reported the presence of a few phenolic acids (such as quinic, caffeic, and chlorogenic acids) and flavonoids (including apigenin, kaempferol, and hesperidin) [16]. However, these approaches yielded only a restricted set of metabolites and did not provide an in-depth characterization of the bulb phytochemistry.

To the best of our knowledge, the present work constitutes the first comprehensive phytochemical characterization focused specifically on the bulbs of *M. neglectum* using HPLC-ESI-qTOF-MS. A total of 72 compounds were annotated, spanning multiple chemical families such as flavonoids, hydroxycinnamic acids, terpenoids, fatty acids, and triterpenoid saponins, thus considerably expanding the current phytochemical knowledge of this species.

The detection of homoisoflavanones and triterpenoid saponins in *M. neglectum* bulbs may also represent chemotaxonomic markers within the Asparagaceae family. These flavonoids—such as muscomin, muscomosin, 3′-Hydroxy-3,9-dihydroeucomin and their isomers—have been reported in previous studies of *M. comosum* [15] and *M. neglectum* [17], reinforcing the phytochemical distinctiveness of the genus *Muscari* and highlighting its relevance for comparative metabolomic work. The presence of homoisoflavanones is a shared derived character of the entire tribe to which Muscari belongs (tribe Hyacintheae), distinguishing it from other tribes within the subfamily Scilloideae [18].

When compared with previous studies on *M. neglectum*, the phenolic and flavonoid contents determined in the present work are considerably higher. For instance, previous studies reported a total phenolic content of approximately 18.2% and a flavonoid content of 0.94% in ethanol extracts of *M. neglectum* flowers [7]. By contrast, the bulb extract analysed here reached 65.5 mg GAE/g DE and 14.3 mg Epi/g DE, respectively. Such differences may be attributed to multiple factors, including the plant part used, as bulbs act as storage organs where secondary metabolites often accumulate more extensively than in aerial tissues. In addition, variations in extraction solvent, methodological approaches, and environmental or geographical conditions may also influence the phytochemical yield. These findings suggest that the bulb of *M. neglectum* represents a particularly rich source of phenolics and flavonoids, underscoring the importance of organ-specific analyses in the phytochemical characterisation of this species.

Most phytochemical and bioactivity studies on *M. neglectum* have focused on flowers, aerial parts, or whole-plant extracts, whereas specific investigations of the bulb remain scarce. This distinction is important, as bulb tissues generally function as storage organs and are therefore expected to accumulate distinct classes of metabolites compared to aerial parts. For instance, extracts from both aerial parts (MAPs) and bulbs (MBs) have been evaluated in rodent models, revealing compositional differences between the two tissues, which were mirrored by variations in their biological effects [8]. In this context, our study provides the first in-depth chemical characterisation of *M. neglectum* bulbs, offering a more targeted view of the metabolite composition of this organ and complementing existing data obtained from other plant parts.

In contrast to earlier studies that have predominantly assessed the antioxidant activity of *M. neglectum* extracts using general assays such as DPPH, ABTS, or FRAP, the present work provides more detailed insights by evaluating activity against specific radical species and enzymatic pathways. Other authors, for example, reported that flower extracts exhibited lower DPPH scavenging capacity compared with the synthetic antioxidant BHT [7]. More recently, a study demonstrated that *M. neglectum* extracts exerted gastroprotective effects in vivo, partly attributed to the presence of phenolic compounds such as caffeic acid, kaempferol, and apigenin [8]. Our findings extend these observations by showing that the bulb extract can scavenge superoxide anion (IC_50_ = 848 mg/L) and hypochlorous acid (IC_50_ = 9.2 mg/L), as well as inhibit xanthine oxidase (IC_50_ = 20.6 mg/L), thereby targeting both free radicals and a key enzyme in oxidative stress pathways. Such activities are likely related to the abundance of flavonoids and hydroxycinnamic acids identified in the extract, suggesting that the bulb represents a distinct and potent source of bioactive antioxidants within the species.

Beyond antioxidant activity, several of the compounds identified—such as flavonoids, hydroxycinnamic acids, and triterpenoid saponins—are known for their anti-inflammatory, gastroprotective, and even neuroprotective properties [19,20,21,22]. This supports the potential pharmacological value of *M. neglectum* bulbs, aligning with their traditional medicinal uses in Mediterranean ethnomedicine.

The compositional profile of *M. neglectum* bulbs reported here provides a plausible chemical basis for the antioxidant and antiradical activities observed. Flavonoids—particularly methoxylated and glycosylated derivatives—are well known for their electron-donating and radical-stabilising properties and therefore likely contribute substantially to both the reducing capacity measured by FRAP/TEAC and the scavenging of reactive species such as superoxide and hypochlorous acid [23,24]. Hydroxycinnamic acids (e.g., caffeic and dihydroferulic acids) are additional contributors to radical scavenging owing to their conjugated phenolic structures, while oxylipin derivatives and certain polyunsaturated fatty acids may also modulate reactivity toward oxygen-derived radicals [25]. Correlations between total phenolic/flavonoid content and antioxidant capacity have been documented in *Muscari* species [7], supporting the proposal that the high TPC/TFC measured in our bulbs underpins at least part of the activity profile. Nonetheless, while these associations are consistent with the literature, definitive attribution of specific activities to individual compounds requires targeted fractionation and bioactivity-guided isolation, as well as quantitative correlation (e.g., Pearson/Spearman analysis) between compound abundance and assay outcomes.

The observed differences among the antioxidant and radical scavenging assays may reflect the distinct reaction mechanisms and selectivity of each method toward different reactive species. The FRAP and TEAC assays, based on single-electron transfer (SET) reactions, evaluate the overall reducing power of the extract. ·O_2_^−^ and HOCl assays involve radical-specific mechanisms that depend more strongly on the structure, polarity, and substitution pattern of the antioxidant molecules.

Phenolic acids and flavonoids bearing ortho-dihydroxyl or methoxy groups on the aromatic ring are particularly effective against oxygen-centered radicals, while more hydrophobic or glycosylated derivatives may exhibit stronger reactivity toward chlorine-based oxidants such as HOCl [26,27]. These structural features modulate hydrogen-donating ability and electron transfer, explaining the disparity observed among the different radical scavenging assays.

To better understand these relationships, a correlation analysis was performed between TPC, TFC, and the various antioxidant parameters. Strong positive correlations were observed between TPC/TFC and both FRAP (r = 0.91) and TEAC (r = 0.88), indicating that phenolic compounds largely account for the reducing capacity of the extract, consistent with previous reports in medicinal and aromatic plants [28,29]. In contrast, weaker correlations were found between TPC/TFC and radical-specific assays (r = 0.63 for ·O_2_^−^ and r = 0.58 for HOCl), suggesting that other non-phenolic metabolites such as saponins, fatty acids, or oxylipins may also contribute to the overall antioxidant performance [26,27].

These results highlight the complexity of antioxidant mechanisms and the complementary nature of different in vitro assays when evaluating the biological relevance of plant extracts.

Beyond their antioxidant potential, the *M. neglectum* bulb extracts also exhibited anti-inflammatory activity, as evidenced by their ability to scavenge ·NO radicals and inhibit LOX. The ·NO radical plays a pivotal role in inflammatory processes by modulating vasodilation and oxidative stress, while LOX catalyzes the oxidation of polyunsaturated fatty acids to generate pro-inflammatory leukotrienes. In the present study, the extract showed moderate ·NO scavenging capacity (IC_50_ = 78 ± 3 mg/L) and LOX inhibition (IC_50_ = 66 ± 2 mg/L), confirming its potential to interfere with both reactive nitrogen species and enzymatic inflammatory pathways. These results are consistent with previous reports demonstrating that phenolic acids and flavonoids, particularly those with ortho-dihydroxyl or methoxy substituents, can attenuate inflammation through ·NO suppression and LOX inhibition [11,30]. Moreover, triterpenoid saponins, which were also detected in the extract, are known to exert anti-inflammatory effects by modulating eicosanoid biosynthesis and reducing the activity of lipoxygenase and cyclooxygenase enzymes [31,32]. Together, these findings suggest that the anti-inflammatory effects of *M. neglectum* bulbs arise from a synergistic interplay between phenolic compounds, flavonoids, and saponins, complementing their antioxidant mechanisms and providing a chemical rationale for the traditional use of this plant in inflammatory disorders.

Collectively, the pharmacological properties of *M. neglectum* bulbs can be attributed mainly to the combined action of phenolic acids, flavonoids, and triterpenoid saponins. Phenolic acids and flavonoids are primarily responsible for the antioxidant and radical scavenging effects through electron transfer and hydrogen donation mechanisms, whereas triterpenoid saponins contribute to anti-inflammatory activity by modulating eicosanoid biosynthesis and inhibiting LOX-mediated pathways. This synergistic interaction among chemical classes provides a coherent explanation for the observed bioactivities and supports the traditional medicinal uses of *M. neglectum*.

In this study, we intentionally focused on the untargeted analysis of the complete *M. neglectum* bulb extract to capture potential synergistic interactions among its compounds, which likely contribute to the observed pharmacological effects. While targeted fractionation could provide detailed information on individual metabolites, our approach prioritizes the bioactivity of the whole extract, which is directly relevant for phytopharmaceutical applications. This strategy allows us to assess the combined effect of all bioactive constituents, reflecting a more holistic view of the extract’s therapeutic potential, and highlights the value of untargeted metabolomics in revealing chemical diversity and functional synergy in plant storage organs. Targeted fractionation may nonetheless represent a valuable future direction, and we note this as a potential avenue for further research.

Importantly, the use of HPLC-ESI-qTOF-MS enabled the detection of minor metabolites, including oxylipins and highly glycosylated saponins, which would likely remain undetected with GC–MS or conventional phytochemical screening methods. This highlights the value of high-resolution untargeted metabolomics for capturing the chemical diversity of storage organs such as bulbs [33].

A subset of the detected molecular features in the *M. neglectum* bulb extract remained unidentified, highlighting the presence of potentially novel metabolites or structurally modified derivatives. While the current study provided a comprehensive annotation of 72 compounds, quantitative profiling of individual constituents was not performed, limiting the ability to directly link specific metabolites to the observed antioxidant and enzyme-inhibitory activities. Moreover, variations in phytochemical composition may arise from factors such as geographic origin, harvest season, and environmental conditions [34,35,36], which could partly account for differences compared with previous reports on *M. neglectum*.

These findings underscore the potential of *M. neglectum* bulbs as a valuable source of bioactive metabolites within the Asparagaceae family. Future studies should employ quantitative targeted analyses of key metabolites, bioactivity-guided fractionation to pinpoint active constituents, and comparative organ-specific investigations across tissues and related *Muscari* species. In vivo models assessing antioxidant, anti-inflammatory, and gastroprotective effects will also be essential to clarify the contributions of specific compounds, explore novel metabolites within the bulb extract, and translate these findings into pharmacological relevance.

## 4. Materials and Methods

### 4.1. Chemicals

All chemicals employed in this study were of analytical reagent grade and utilised as supplied. Acetonitrile and formic acid of LC-MS grade, intended for use in mobile phases, were obtained from Riedel-de-Haën (Honeywell, Charlotte, NC, USA). Ultrapure water for solution preparation was produced using a Milli-Q system (Millipore, Bedford, MA, USA), and absolute ethanol was procured from VWR Chemicals (Radnor, PA, USA).

Acetic acid, sodium carbonate, sodium hydroxide, sodium nitrite, TPTZ (2,4,6-tris(2-pyridyl)-s-triazine and hydrochloric acid were purchased from Fluka (Honeywell, NC, USA). Folin reagent, gallic acid (GA), sodium phosphate monobasic and dibasic, potassium persulfate, ABTS (2,2-azinobis (3-ethylbenzothiazoline-6-sulphonate)), Trolox (6-hydroxy-2,5,7,8-tetramethylchroman-2-carboxylic acid), Tris (tri(hydroxymethyl)aminomethane), sodium acetate, heptahydrate ferrous sulphate, aluminium chloride hexahydrate, ferric chloride, DHR (dihydrorhodamine), potassium dihydrogen phosphate anhydrous, NBT (nitrotetrazolium blue chloride), DAF-2 (diaminofluorescein diacetate) and Cayman’s xanthine oxidase fluorometric assay kit were purchased from Sigma-Aldrich (St. Louis, MO, USA). NOC-5 was purchased from Chemcruz (Santa Cruz Biotechnology, Dallas, TX, USA).

### 4.2. Extraction Process of Plant Bulbs

Specimens of *M. neglectum* were collected from a wild population in the northern part of the city of Granada, southern Spain, in October 2024, at an early developmental stage when bulbs were beginning to germinate (as illustrated in Appendix A). The bulbs were subsequently isolated, peeled, and thoroughly washed with distilled water. To accelerate the drying process, the bulbs were then manually fragmented. Air-drying was carried out in a cool, dark, and well-ventilated environment for seven days at a constant temperature of 25 °C. Following desiccation, the dried bulbs were ground using a basic mill A 10 (IKA-Werke GmbH & Co., Staufen, Germany) until a fine, homogeneous powder with a particle size of less than 1 mm was obtained, thereby facilitating subsequent extraction procedures.

The extraction process was performed using a solid–liquid extraction technique. Precisely 2 g of the powdered plant material was weighed into dark glass vials, to which 20 mL of an ethanol–water mixture (80:20, *v*/*v*) was added. The vials were then placed in an incubator with orbital shaking, set at 45 °C and 150 rpm, for a period of 2 h to ensure adequate mixing and solvent penetration.

Preliminary extractions were performed using different ethanol–water ratios (100:0, 80:20, 50:50, 20:80, 0:100, *v*/*v*) under the same solid–liquid extraction conditions. The TPC of the resulting extracts was determined using the Folin–Ciocalteu assay to assess the efficiency of phenolic compound recovery. Based on these results, the 80:20 ethanol–water mixture was selected for all subsequent experiments, as it preserves a substantial fraction of total phenolics while favouring the extraction of polar phenolic compounds of biological relevance. This extract was then used for all further phytochemical characterization and bioactivity analyses.

The conditions employed for obtaining the extract proved suitable, preserving the integrity of the target compounds. The extraction protocol was based on previously established methodologies, with targeted modifications implemented to optimise its application to this particular plant matrix.

### 4.3. HPLC-ESI-qTOF-MS Analysis

Chromatographic analysis was conducted in reversed-phase mode using a C18 ACQUITY UPLC BEH column (1.7 µm, 2.1 mm × 150 mm, 130 Å; Waters Corporation, Milford, MA, USA). The column temperature was maintained at 60 °C throughout the analysis. The mobile phases consisted of (A) water acidified with 0.1% formic acid (*v*/*v*) and (B) acetonitrile. The optimised mobile phase gradient applied for compound separation was as follows: 0.00 min [A:B 100:0], 5.00 min [A:B 90:10], 18.00 min [A:B 15:85], 24.00 min [A:B 0:100], 25.50 min [A:B 0:100], 26.50 min [A:B 95:5], and 32.50 min [A:B 95:5]. A constant mobile phase flow rate of 0.4 mL/min was employed, with an injection volume of 5 µL.

Mass spectrometric data acquisition was performed in electrospray ionisation (ESI) negative mode over a mass range of 50–1200 *m*/*z*. The instrument parameters were set as follows: drying gas flow rate, 10 L/min; drying gas temperature, 200 °C; nebuliser pressure, 20 psig; sheath gas temperature, 350 °C; sheath gas flow rate, 12 L/min; capillary voltage (VCap), 4000 V; and nozzle voltage, 500 V.

Chromatographic and mass spectrometric conditions and parameters were applied in accordance with a previously optimised method. The acquired raw data were initially converted using MSConvertGUI software and subsequently processed in MZmine version 4.3.0. This data processing workflow comprised several sequential steps, including background noise detection, ADAP chromatogram builder, ADAP chromatogram deconvolution, isotope grouping, and alignment. The specific parameters employed during data processing have been detailed in a previous publication [37].

Subsequent prediction of molecular formulae and chemical structures was performed using Sirius version 5.8.1. The information generated by Sirius was compared against various databases, including the Human Metabolome Database (HMDB), MassBank of North America, and CEU Mass Mediator for compound annotation purposes. Identified compounds were classified according to four confidence levels: Level 1 identification was achieved through comparison with commercial standards; Level 2 was based on the comparison of experimental MS/MS spectra with those available in the referenced databases; Level 3 involved molecular formula prediction and MS1 spectrum comparison; and Level 4 comprised features that could not be annotated and were thus considered unknown [38].

### 4.4. In Vitro Assays

The assays detailed in the following sections were adapted for implementation in a 96-well polystyrene microplate format, enabling the simultaneous processing of multiple replicates and experimental conditions. Absorbance and fluorescence measurements were subsequently performed using a Synergy H1 monochromator-based multimode microplate reader (BioTek Instruments Inc., Winooski, VT, USA). This instrument allows for high-sensitivity detection and flexible wavelength selection, thereby ensuring precise and reliable acquisition of optical density and fluorescence emission data under the specified assay conditions.

#### 4.4.1. Total Phenolic Content, Total Flavonoid Content and Antioxidant Capacity Measurements

The total phenolic content (TPC) in extracts obtained from *M. neglectum* bulbs was quantified utilising the Folin–Ciocalteu reagent assay, with gallic acid serving as the calibration standard [39]. Results are reported as milligrams of gallic acid equivalents per gram of dry extract (mg GAE/g DE). Each determination was performed in triplicate to guarantee precision and reproducibility. The determination of Total Flavonoid Content (TFC) was conducted following a previously described protocol with minor modifications [40]. Epicatechin was employed as the reference standard, and absorbance readings were recorded at 510 nm. The results were expressed as milligrams of epicatechin equivalents per gram of dry extract (mg EE/g DE).

The antioxidant potential of these bulb extracts was evaluated through two complementary assays: ferric reducing antioxidant power (FRAP) and trolox equivalent antioxidant capacity (TEAC). The FRAP assay followed the protocol established by [41], expressing antioxidant capacity as millimoles of ferrous sulphate equivalents per gram of dry extract (mmol FeSO_4_/g DE). The TEAC assay, which measures the ability to quench the ABTS radical cation, was conducted as described by [42], with outcomes expressed in micromoles of Trolox equivalents per gram of dry extract (μmol Trolox/g DE).

#### 4.4.2. Evaluation of Free Radical and ROS Scavenging Potential

The inhibitory capacity of *M. neglectum* bulb extract against reactive species, including ·O_2_^−^, HOCl and ·NO, was investigated. The assays were conducted following previously established protocols, with minor adaptations [13]. The superoxide radical scavenging activity was evaluated colorimetrically by monitoring the reduction in nitroblue tetrazolium (NBT) to diformazan purple. Inhibition of HOCl was assessed by measuring the oxidation of dihydrorhodamine (DHR) to rhodamine. Nitric-oxide anions were generated by the presence of NOC-5, and 4,5-diaminofluorescein (DAF-2) was used as the fluorescent probe applied. Epicatechin (EPI) and gallic acid (GA) served as positive controls for the inhibition of ·O_2_^−^, HOCl and ·NO radicals. Results were expressed as IC_50_ values, calculated from concentration-response curves of the *M. neglectum* bulb extract, with all experiments performed in triplicate to ensure reproducibility.

#### 4.4.3. Xanthine Oxidase Assay

The inhibitory effect of the *M. neglectum* bulb extract on xanthine oxidase (XO) activity was assessed using the Cayman Xanthine Oxidase Fluorometric Assay Kit (Cayman Chemical, Ann Arbor, MI, USA). Briefly, 50 μL of the test extract at various concentrations (10–500 μg/mL) was mixed with 50 μL of XO enzyme solution in a 96-well microplate. The reaction was initiated by adding 50 μL of the xanthine substrate and incubating at 25 °C for 30 min. Fluorescence intensity, corresponding to uric acid formation, was measured at excitation/emission wavelengths of 535/587 nm using a Synergy H1 multimode microplate reader (BioTek Instruments Inc., Winooski, VT, USA). Inhibition (%) was calculated using the formula:(1)$$Inhibition\left(\%\right)\;=\;\frac{A_{control}\;-\;A_{sample}}{A_{control}}\;\times \;100\;$$
where A_control_ is the fluorescence of the enzyme without inhibitor and A_sample_ is the fluorescence in the presence of the extract. Epicatechin (EPI) was used as the positive control to validate the assay.

Standard xanthine oxidase inhibitors such as allopurinol were not included in this study because the primary objective was to evaluate the inhibitory potential of novel natural extracts. Future studies will include well-characterized inhibitors to allow direct comparison and better contextualization of biological significance.

#### 4.4.4. Evaluation of Lipoxygenase Inhibitor Assay

The LOX inhibitory activity of the *M. neglectum* bulb extract was evaluated using the LOX Inhibitor Screening Assay Kit (Cayman Chemical, Ann Arbor, MI, USA), following the manufacturer’s instructions.

The assay is based on the spectrophotometric detection of hydroperoxides formed during the LOX-catalyzed oxidation of arachidonic acid, which serves as the enzyme substrate. Briefly, the reaction mixture contained 10 µL of sample solution (dissolved in assay buffer), 90 µL of lipoxygenase enzyme solution, and 100 µL of arachidonic acid substrate solution. The formation of the hydroperoxide product was monitored by measuring absorbance at 490 nm using a microplate reader (BioTek Synergy HTX, Winooski, VT, USA). The *M. neglectum* bulb extract was tested at concentrations ranging from 10 to 200 mg/L to obtain the inhibition curve. NDGA, a well-known lipoxygenase inhibitor, was used as a positive control, while the assay buffer served as a negative control.

The percentage of inhibition was calculated according to the equation:(2)$$Inhibition\left(\%\right)\;=\;\left(1\;-\;\frac{A_{sample}\;-\;A_{blank}}{A_{control}\;-\;A_{blank}}\right)\;\times \;100\;$$
where A_sample_ represents the absorbance of the sample reaction, A_control_ represents the absorbance of the control reaction without inhibitor, and A_blank_ represents the background absorbance. The concentration of extract required to inhibit 50% of enzyme activity (IC_50_) was determined by non-linear regression

#### 4.4.5. Statistical Analysis

All experiments were carried out in triplicate (*n* = 3), and results are expressed as mean ± standard deviation (SD). The correlation between total phenolic content (TPC), total flavonoid content (TFC), and antioxidant activities was analyzed using Pearson’s correlation coefficient (r). Statistical significance was set at *p* < 0.05. All analyses were performed using GraphPad Prism 9.0 (GraphPad Software Inc., San Diego, CA, USA).

## 5. Conclusions

This study offers the first thorough phytochemical characterisation of *M. neglectum* bulbs using HPLC-ESI-qTOF-MS. A total of 72 compounds were identified, including fatty acids, iridoid glycosides, triterpenoid saponins, hydroxycinnamic acids, and flavonoids. Notably, several homoisoflavanones such as muscomin and muscomosin isomers, together with related derivatives, were detected; these metabolites are considered characteristic of the *Muscari* genus and reinforce its chemotaxonomic distinctiveness within the Asparagaceae family, subfamily Scilloideae. Due in large part to the abundance of flavonoids and hydroxycinnamic acids, the bulb extract demonstrated significant antioxidant and radical scavenging activities, including superoxide and hypochlorous acid scavenging, as well as xanthine oxidase inhibition. The significance of organ-specific analyses was highlighted by the bulbs′ higher total phenolic and flavonoid contents when compared to earlier research on flowers or aerial tissues. A subset of molecular features remained unidentified, indicating the presence of potentially novel metabolites that warrant further investigation. In addition, the extract exhibited measurable anti-inflammatory activity, as evidenced by its ability to scavenge nitric oxide radicals and inhibit lipoxygenase activity. These findings suggest that both phenolic compounds and triterpenoid saponins contribute to the modulation of oxidative and enzymatic inflammatory pathways, providing a biochemical basis for the traditional use of *Muscari* species in the treatment of inflammatory disorders. Overall, these findings establish *M. neglectum* bulbs as a rich reservoir of bioactive compounds and underscore their potential relevance for future pharmacological and functional studies within the Asparagaceae family.

## Figures and Tables

**Figure 1 molecules-30-04351-f001:**
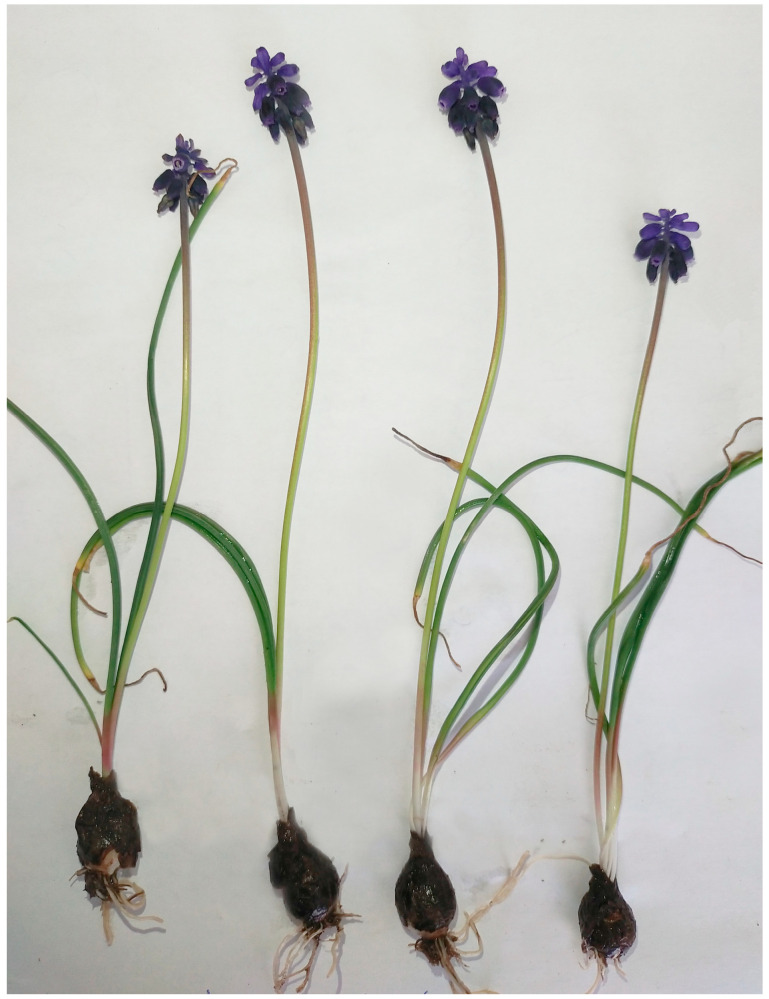
Complete specimens of *Muscari neglectum*.

**Table 1 molecules-30-04351-t001:** Effect of ethanol–water ratio on the total phenolic content of *M. neglectum* bulb extracts.

Hydroalcoholic Mixture (EtOH:H_2_O, *v*/*v*)	100:0	80:20	50:50	20:80	0:100
**TPC** **(mg GAE/g DE)**	96.3 ± 0.8	65.5 ± 0.3	30.4 ± 0.8	10.4 ± 0.2	5.9 ± 0.6

TPC: Total Polar Compounds; GAE: Gallic Acid Equivalent; DE: Dry Extract. Data are means ± standard deviation (*n* = 3).

**Table 2 molecules-30-04351-t002:** Identification of phytochemical compounds in *M. neglectum* bulbs extract by HPLC-ESI-qTOF-MS.

Peak	Rt (min)	[M − H]^−^ Experimental	[M − H]^−^ Theoretical	Error (ppm)	Mol. Formula	Proposed Compound	MS/MS Fragments	Level of Annotation	References	Chemical Family
1	0.729	218.1032	218.1034	−0.90	C_9_H_17_NO_5_	Pantothenic Acid	152	2	HMDB0000210	Vitamin (B-complex, water-soluble)
2	0.928	131.0458	131.0462	−3.17	C_4_H_8_N_2_O_3_	Asparagine	104; 113	2	HMDB0033780	Amino acid
3	0.988	387.1144	387.1144	−0.04	C_13_H_24_O_13_	Unknown	89; 179	4	-	-
4	1.121	133.0147	133.0142	3.41	C_4_H_6_O_5_	Malic acid isomer 1	115; 71	2	HMDB0000156	Organic acid (α-hydroxy acid)
5	1.253	133.0141	133.0142	−1.11	C_4_H_6_O_5_	Malic acid isomer 2	115; 71	2	HMDB0000156	Organic acid (α-hydroxy acid)
6	3.011	189.0772	189.0768	1.87	C_8_H_14_O_5_	3-Hydroxysuberic acid	59; 129	2	HMDB0000325	Organic acid (dicarboxylic acid)
7	5.161	195.0662	195.0663	−0.42	C_10_H_12_O_4_	Dihydroferulic acid	165; 133; 59	2	HMDB0062121	Hydroxycinnamic acid
8	6.696	179.0344	179.0350	−3.25	C_9_H_8_O_4_	Caffeic acid	135; 134	2	HMDB0001964	Hydroxycinnamic acid
9	7.845	369.0841	369.0827	3.74	C_16_H_18_O_10_	Fraxin	85; 223; 129	2	HMDB0252486	Coumarin (glycosylated)
10	8.424	433.2093	433.2079	3.18	C_20_H_34_O_10_	Monoterpene derivative	387; 89	2	[12]	Terpenoid (monoterpene)
11	8.832	301.1086	301.1081	1.50	C_17_H_18_O_5_	Isomucronulatol	135	2	PubChem: 602152	Flavonoid (isoflavonoid derivative)
12	9.234	329.0645	329.0667	−6.61	C_17_H_14_O_7_	Dimethylquercetin	223; 286 ^a^	2	HMDB0029263	Flavonoid (flavonol)
13	9.405	725.1895	725.1935	−5.45	C_32_H_38_O_19_	Kaempferol 3-(2G-apiosylrobinobioside)	269	2	HMDB0039759	Flavonoid (glycosylated flavonol)
14	9.565	829.2356	829.2408	−6.27	C_36_H_46_O_22_	Sesaminol 2-*O*-triglucoside	621; 783	2	HMDB0041775	Lignan (glycosylated)
15	10.182	931.2736	931.2725	1.19	C_40_H_52_O_25_	Torososide B	271	2	HMDB0034855	Iridoid (glycoside)
16	10.309	467.2149	467.2134	3.21	C_20_H_36_O_12_	Unknown	-	4	-	-
17	10.535	491.2131	491.2134	−0.61	C_22_H_36_O_12_	Jasminoside S	445; 194; 209	2	PubChem: 71552547	Iridoid (glycoside)
18	10.595	317.0672	317.0667	1.65	C_16_H_14_O_7_	3,3′,5,7-Tetrahydroxy-4′-methoxyflavanone	195; 167	2	FDB016577	Flavonoid (flavanone)
19	10.711	621.182	621.1825	−0.80	C_29_H_34_O_15_	Pectolinarin	313; 125	2	HMDB0256193	Flavonoid (glycosylated flavonoid)
20	11.092	331.0832	331.0823	2.64	C_17_H_16_O_7_	3,4′,5-Trihydroxy-3′,7-dimethoxyflavanone isomer 1	301; 271; 286	2	HMDB0037503	Flavonoid (methoxylated flavanone)
21	11.466	1105.5455	1105.5436	1.69	C_53_H_86_O_24_	Unknown	-	4	-	-
22	11.527	301.0717	301.0718	−0.21	C_16_H_14_O_6_	Hesperetin isomer 1	195; 167	2	PubChem: 72281	Flavonoid (flavanone)
23	11.665	301.0704	301.0718	−4.52	C_16_H_14_O_6_	Hesperetin isomer 2	195; 167	2	PubChem: 72281	Flavonoid (flavanone)
24	11.803	331.0804	331.0823	−5.82	C_17_H_16_O_7_	3,4′,5-Trihydroxy-3′,7-dimethoxyflavanone isomer 2	301; 271; 286	2	HMDB0037503	Flavonoid (methoxylated flavanone)
25	11.935	331.0833	331.0823	2.94	C_17_H_16_O_7_	3,4′,5-Trihydroxy-3′,7-dimethoxyflavanone isomer 3	301; 271; 286	2	HMDB0037503	Flavonoid (methoxylated flavanone)
26	12.018	299.0553	299.0561	−2.72	C_16_H_12_O_6_	Luteolin 7-methyl ether	151; 179; 107	2	HMDB0037339	Flavonoid (methylated flavone)
27	12.111	331.0828	331.0823	1.43	C_17_H_16_O_7_	3,4′,5-Trihydroxy-3′,7-dimethoxyflavanone isomer 4	301; 271; 286	2	HMDB0037503	Flavonoid (methoxylated flavanone)
28	12.321	301.0695	301.0718	−7.51	C_16_H_14_O_6_	Hesperetin isomer 3	195; 167	2	72281	Flavonoid (flavanone)
29	12.42	1119.5232	1119.5323	−8.11	C_64_H_80_O_17_	Unknown	1073	4	-	-
30	12.464	1119.5214	1119.5323	−9.71	C_64_H_80_O_17_	Unknown	1073	4	-	-
31	12.618	329.2334	329.2333	0.16	C_18_H_34_O_5_	9,12,13-trihydroxyoctadec-10-enoic acid	125; 211	2	HMDB0004708	Fatty acid (oxylipin derivative)
32	12.651	345.0983	345.0980	0.94	C_18_H_18_O_7_	Muscomin isomer 1	330; 229; 133 ^b^	2	C00057710	Flavonoid (homoisoflavanones)
33	12.806	811.4232	811.4122	13.61	C_41_H_64_O_16_	Unknown	765; 677	2	-	-
34	13.07	313.0722	313.0718	1.40	C_17_H_14_O_6_	Muscomosin isomer 1	298 ^c^	2	CHEM027488	Flavonoid (homoisoflavanones)
35	13.125	267.1039	267.1027	4.61	C_17_H_16_O_3_	Eugenyl benzoate	135; 93	2	HMDB0032056	Phenylpropanoid ester
36	13.23	315.0871	315.0874	−0.99	C_17_H_16_O_6_	3′-Hydroxy-3,9-dihydroeucomin	116; 144	2	CAS: 107585-75-1	Flavonoid (homoisoflavanones)
37	13.307	345.0972	345.0980	−2.25	C_18_H_18_O_7_	Muscomin isomer 2	330; 229; 133	2	C00057710	Flavonoid (homoisoflavanones)
38	13.379	313.072	313.0718	0.76	C_17_H_14_O_6_	Muscomosin isomer 2	298	2	CHEM027488	Flavonoid (homoisoflavanones)
39	13.434	285.0777	285.0768	2.99	C_16_H_14_O_5_	4′-demethyl-3,9-dihydroeucomin	179	2	[13]	Flavonoid (homoisoflavanones)
40	13.495	313.0731	313.0718	4.27	C_17_H_14_O_6_	Muscomosin isomer 3	298	2	CHEM027488	Flavonoid (homoisoflavanones)
41	13.66	583.1971	583.1974	−0.44	C_34_H_32_O_9_	5′,7-Dibenzyloxy-3′,4′,5,6,8-pentamethoxyflavone isomer 1	327; 553	2	PubChem: 129671223	Flavonoid (highly methoxylated flavone)
42	13.814	509.2005	-	-	-	Unknown	-	4	-	-
43	13.859	815.4426	-	-	-	Unknown	-	4	-	-
44	13.947	1105.5401	-	-	-	Unknown	-	4	-	-
45	14.101	422.328	422.3276	0.99	C_25_H_45_NO_4_	O-Linoleoylcarnitine	380	2	HMDB0240780	Fatty acid conjugate (carnitine ester)
46	14.2	293.1765	293.1758	2.28	C_17_H_26_O_4_	Gingerol	221; 236	2	HMDB0005783	Phenolic ketone
47	14.355	553.1867	553.1868	−0.16	C_33_H_30_O_8_	5-hydroxy-6-(7-hydroxy-5,8-dimethoxy-2-phenyl-chroman-4-yl)-7-methoxy-2-phenyl-chroman-4-one isomer 1	205; 431	2	CNP0301559.0	Flavonoid (complex flavonoid dimer)
48	14.421	583.199	583.1974	2.82	C_34_H_32_O_9_	5′,7-Dibenzyloxy-3′,4′,5,6,8-pentamethoxyflavone isomer 2	327; 553	2	PubChem: 129671223	Flavonoid (highly methoxylated flavone)
49	14.63	553.1863	553.1868	−0.89	C_33_H_30_O_8_	5-hydroxy-6-(7-hydroxy-5,8-dimethoxy-2-phenyl-chroman-4-yl)-7-methoxy-2-phenyl-chroman-4-one isomer 2	205; 431	2	CNP0301559.0	Flavonoid (complex flavonoid dimer)
50	14.779	327.0858	327.0874	−4.93	C_18_H_16_O_6_	5-Hydroxy-4′,7,8-trimethoxyflavone	165; 139	2	HMDB0037458	Flavonoid (methoxylated flavone)
51	14.851	493.2025	493.2020	0.92	C_32_H_30_O_5_	Unknown	-	4	-	-
52	15.165	565.1863		-		Unknown	-	4	-	-
53	15.341	537.1898	537.1919	−3.87	C_33_H_30_O_7_	7-(Benzyloxy)-3-(3-(benzyloxy)-4-methoxybenzyl)-5,6-dimethoxychroman-4-one	205; 337	2	[14]	Flavonoid (benzylated flavonoid)
54	15.44	565.1901	-	-	-	Unknown	-	4	-	-
55	15.529	493.2021	493.2020	0.11	C_32_H_30_O_5_	Unknown	-	-	-	-
56	16.074	579.2024	579.2024	−0.07	C_35_H_32_O_8_	Officinin A isomer 1	353; 93	2	C00056992	Triterpenoid saponin
57	16.273	579.202	579.2024	−0.76	C_35_H_32_O_8_	Officinin A isomer 2	353; 93	2	C00056992	Triterpenoid saponin
58	16.565	519.3324	519.3327	−0.63	C_30_H_48_O_7_	Cucurbitacin P	-	3	PubChem: 441822	Triterpenoid (cucurbitacin)
59	18.687	301.2178	301.2173	1.65	C_20_H_30_O_2_	Eicosapentaenoic Acid	59	2	PubChem: 5282847	Polyunsaturated fatty acid (PUFA, omega-3)
60	19.161	327.2336	-	-	-	Unknown	-	4	-	-
61	19.596	279.2333	279.2330	1.24	C_18_H_32_O_2_	Linoleic acid	-	3	HMDB0000673	Polyunsaturated fatty acid (PUFA, omega-6)
62	19.596	379.1591	379.1551	10.56	C_23_H_24_O_5_	Garcinone A	-	3	HMDB0029509	Xanthone derivative
63	20.329	355.1577	355.1551	7.33	C_21_H_24_O_5_	Gingerenone A	311	2	HMDB0035403	Diarylheptanoid
64	20.451	381.1748	381.1707	10.63	C_23_H_26_O_5_	3-Oxo-2,4-bis(3-phenylpropyl)pentanedioic acid	337	2	PubChem: 69563854	Dicarboxylic acid derivative
65	20.958	619.4217	619.4215	0.26	C_36_H_60_O_8_	Ginsenoside Rh4 isomer 1 ^d^	574	2	HMDB0041534	Triterpenoid saponin (ginsenoside)
66	21.14	619.4218	619.4215	0.42	C_36_H_60_O_8_	Ginsenoside Rh4 isomer 2	574	2	HMDB0041534	Triterpenoid saponin (ginsenoside)
67	21.3	621.4371	621.4372	−0.15	C_36_H_62_O_8_	(20*R*)-Ginsenoside Rh2 isomer 1	-	3	HMDB0039544	Triterpenoid saponin (ginsenoside)
68	21.492	621.4383	621.4372	1.78	C_36_H_62_O_8_	(20*R*)-Ginsenoside Rh2 isomer 2	-	3	HMDB0039544	Triterpenoid saponin (ginsenoside)
69	21.625	383.1901	-	-	-	Unknown	-	4	-	-
70	21.911	758.5442	-	-	-	Unknown	-	4	-	-
71	23.157	961.6126	-	-	-	Unknown	-	4	-	-
72	25.445	653.2629	653.2662	−5.09	C_28_H_46_O_17_	Unknown	593	4	-	-

Rt: Retention Time; Mol. Formula: Molecular Formula. ^a.^Mass fragmentation in Appendix A. ^b.^Mass fragmentation in Appendix A. ^c.^Mass fragmentation in Appendix A. ^d.^Mass fragmentation in Appendix A.

**Table 3 molecules-30-04351-t003:** Evaluation of total phenolic content, total flavonoid content, and antioxidant capacity.

Sample	TPC (mg GAE/g DE)	TFC (mg Epi/g DE)	FRAP (mmol Fe^2+^/g DE)	TEAC (mmol TE/g DE)
*M. neglectum* bulbs	65.5 ± 0.3	14.3 ± 0.3	0.26 ± 0.02	0.45 ± 0.02

TPC: Total Polar Compounds; TFC: Total Flavonoid Content; FRAP: Ferric Reducing Antioxidant Power Assay; TEAC: Trolox Equivalent Antioxidant Capacity; Epi: Epicatechin; GAE: Gallic Acid Equivalent; DE: Dry Extract; TE: Trolox Equivalent. Data are means ± standard deviation (*n* = 3).

**Table 4 molecules-30-04351-t004:** Evaluation of radical scavenging.

Sample	·O_2_^−^ (mg/L) ^1^	HOCl (mg/L) ^1^
*M. neglectum* bulbs	848 ± 20	9.2 ± 0.9
Gallic acid	50 ± 3	3.8 ± 0.3
Epicatechin	60 ± 1	0.18 ± 0.01

Data are means ± standard deviation (*n* = 3). ^1^ Inhibitory Concentration at 50%.

**Table 5 molecules-30-04351-t005:** Evaluation of xanthine oxidase assay.

Sample	XOD (mg/L) ^1^
*M. neglectum* bulbs	20.6 ± 0.4
Epicatechin	9 ± 1

Data are means ± standard deviation (*n* = 3). ^1^ Inhibitory Concentration at 50%.

**Table 6 molecules-30-04351-t006:** Evaluation of radical scavenging.

Sample	·NO (mg/L) ^1^
*M. neglectum* bulbs	78 ± 3
Gallic acid	14.2 ± 0.8
Epicatechin	8.7 ± 0.2

Data are means ± standard deviation (*n* = 3). ^1^ Inhibitory Concentration at 50%.

**Table 7 molecules-30-04351-t007:** Evaluation of lipoxygenase inhibitor assay.

Sample	LOX (mg/L) ^1^
*M. neglectum* bulbs	66 ± 2
NGDA	3.63 ± 0.02

NDGA: Nordihydroguaiaretic acid; Data are means ± standard deviation (*n* = 3). ^1^ Inhibitory Concentration at 50%.

## Data Availability

The data presented in this study are available on request from the corresponding author.

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
