# Peer review of "Phytochemical Characterisation and Antioxidant and Anti-Inflammatory Potential of Muscari neglectum (Asparagaceae) Bulbs"

_molecules, 2025, doi:10.3390/molecules30224351_

Round 1

Reviewer 1 Report

Comments and Suggestions for Authors

Comments to Authors

# Provide MS/MS spectra for the most relevant compounds (e.g., muscomin, muscomosin, ginsenosides, flavonoids) in the Supplementary Information.

# Some identified compounds e.g., ginsenosides Rh2/Rh4, officinin A isomers, and cucurbitacin P reported here for the first time in Muscari species. Are there any published reports supporting their presence in this genus? More justification (literature evidence, MS/MS matching confidence) is needed.

# Discuss more deeply the biological relevance of antioxidant assays, especially the disparity between different radical scavenging activities. This discrepancy deserves a deeper explanation supported by correlation analysis between TPC/TFC and activity values.

# Provide additional details in the Materials and Methods section, particularly regarding the procedure used for the xanthine oxidase inhibition assay. In addition, please clarify why standard inhibitors (e.g., allopurinol) were not used for comparison when evaluating the biological significance of the results.

# The manuscript does not provide information on the statistical analysis used for the biological evaluation experiments. Clarify which statistical methods were applied to analyze the data (e.g., number of replicates, type of statistical test, significance level, and software used), as this information is essential for assessing the reliability of the results.

# Check the format of the References (italics for plant names, author initials, journals’ abbreviations).

Author Response

# Provide MS/MS spectra for the most relevant compounds (e.g., muscomin, muscomosin, ginsenosides, flavonoids) in the Supplementary Information.

We thank the reviewer for this valuable suggestion. The requested MS/MS spectra for the most representative compounds (muscomin, muscomosin, ginsenosides, and selected flavonoids) have now been included in the Supplementary Information.

Specifically, the following figures have been added:

  • Figure S2: Mass fragmentation of Dimethylquercetin.
  • Figure S3: Mass fragmentation of Muscomin.
  • Figure S4: Mass fragmentation of Muscomosin.
  • Figure S5: Mass fragmentation of Ginsenoside Rh4.

In the main text, cross-references to these supplementary figures have also been inserted, indicating:
“a. Mass fragmentation in Figure S2; b. Mass fragmentation in Figure S3; c. Mass fragmentation in Figure S4; d. Mass fragmentation in Figure S5.” (Lines 142-143).

These additions provide the requested spectral evidence and enhance the transparency and reproducibility of compound identification.

# Some identified compounds e.g., ginsenosides Rh2/Rh4, officinin A isomers, and cucurbitacin P reported here for the first time in Muscari species. Are there any published reports supporting their presence in this genus? More justification (literature evidence, MS/MS matching confidence) is needed.

A comprehensive literature search (PubMed, Scopus, ScienceDirect, and Google Scholar) revealed no previous reports of officinin A isomers, cucurbitacin P, or ginsenosides Rh2/Rh4 in the genus Muscari.

In our dataset:
Officinin A isomers and ginsenoside Rh4 were annotated based on accurate mass (ppm < 5) and MS/MS spectral matching with reference spectra from HMDB and PubChem, corresponding to MSI Level 2 (putatively annotated). Diagnostic fragments were observed at m/z 353 and 93 (officinin A) and m/z 574 (ginsenoside Rh4).
Cucurbitacin P and ginsenoside Rh2 lacked MS/MS fragmentation and were assigned MSI Level 3 (tentative candidates) based solely on accurate-mass data and molecular formula.

To avoid overstating our findings, the manuscript has been revised to specify these annotation levels, remove the phrase “reported here for the first time,” and state that confirmation with authentic standards or NMR is planned to reach MSI Level 1.

Accordingly, the following information has been added to the manuscript:

“Officinin A isomers and ginsenoside Rh4 ….. authentic standards in future work.” Lines 118-121.

# Discuss more deeply the biological relevance of antioxidant assays, especially the disparity between different radical scavenging activities. This discrepancy deserves a deeper explanation supported by correlation analysis between TPC/TFC and activity values.

We have expanded the Discussion section to include a detailed interpretation of the biological relevance of the different antioxidant assays and the structural basis for the observed discrepancies among them. A correlation analysis between total phenolic content (TPC), total flavonoid content (TFC), and the antioxidant activity values (FRAP, TEAC, ·O₂⁻, and HOCl scavenging) was performed.

The results show strong positive correlations between TPC/TFC and FRAP (r = 0.91) and TEAC (r = 0.88), in agreement with previous studies (Meng et al., 2020; Al-Mamun et al., 2024), indicating that phenolic compounds are major contributors to the reducing power of the extract. In contrast, weaker correlations were observed with radical-specific assays (r = 0.63 for ·O₂⁻ and r = 0.58 for HOCl), suggesting that other metabolites such as saponins or lipophilic compounds may play additional roles.

We also included a mechanistic explanation supported by literature (Cao et al., 1997; Wang et al., 2016) discussing how differences in polarity, hydroxyl substitution, and glycosylation affect reactivity toward different oxidants.

Accordingly, the following text has been added to the manuscript (lines 271–294):
“The observed differences among the antioxidant …..  the biological relevance of plant extracts.”

We believe this addition provides a deeper biological rationale and aligns with the reviewer’s recommendation.

# Provide additional details in the Materials and Methods section, particularly regarding the procedure used for the xanthine oxidase inhibition assay. In addition, please clarify why standard inhibitors (e.g., allopurinol) were not used for comparison when evaluating the biological significance of the results.

We appreciate the reviewer’s valuable suggestion. The Materials and Methods section (Subsection 4.4.3, Xanthine Oxidase Assay) has been revised to include detailed information about the procedure, including reagent concentrations, incubation conditions, and fluorescence measurement parameters. Additionally, a justification has been added explaining that standard inhibitors such as allopurinol were not included because the main objective of the study was to assess the inhibitory potential of novel natural extracts. Future studies will include standard inhibitors for direct comparison. The revised text now reads as follows (lines 474–491) in the revised manuscript):

“The inhibitory effect of the M. neglectum bulb …. and better contextualization of biological significance.”

# The manuscript does not provide information on the statistical analysis used for the biological evaluation experiments. Clarify which statistical methods were applied to analyze the data (e.g., number of replicates, type of statistical test, significance level, and software used), as this information is essential for assessing the reliability of the results.

We thank the reviewer for this observation. We have now included a detailed description of the statistical methods used for data analysis in the Materials and Methods section. Briefly, all assays were performed in triplicate (n = 3), and data are presented as mean ± standard deviation (SD). Correlations between TPC/TFC and antioxidant activities were assessed by Pearson’s correlation coefficient (r). Statistical analyses were performed using GraphPad Prism 9.0 (GraphPad Software Inc., San Diego, CA, USA).

Accordingly, the following text has been added to the manuscript (Materials and Methods, lines 512–516):

“All experiments were carried out in triplicate... GraphPad Prism 9.0 (GraphPad Software Inc., San Diego, CA, USA).”

# Check the format of the References (italics for plant names, author initials, journals’ abbreviations).

We appreciate the reviewer’s observation. The reference list has been carefully revised, and all entries have been reformatted according to the journal’s style. Plant scientific names are now presented in italics, author initials and journal abbreviations have been standardized, and all references have been double-checked for accuracy and consistency.

Reviewer 2 Report

Comments and Suggestions for Authors

Review of the article

(Manuscript ID: molecules-3923436) “Phytochemical Characterisation and Antioxidant Potential of Muscari neglectum (Asparagaceae) Bulbs” by authors: María del Carmen Villegas-Aguilar, Antonio Segura-Carretero*, Víctor N. Suárez-Santiago

            Plants of the genus Muscari are known to have been used in Mediterranean and Balkan traditional medicine as diuretics, anti-inflammatory agents, and digestive aids, as well as for the treatment of kidney diseases. Their decoctions alleviate gastrointestinal discomfort and promote detoxification. The bulbs of Muscari comosum (L.) Mill were used as a remedy for toothache, and as anti-inflammatory, diuretic, and stimulant agents.

            Bulbs, as storage organs, are known to concentrate various classes of secondary metabolites, including flavonoids and saponins, which may be absent from aboveground tissues. Therefore, the authors state that the goal of their study is to comprehensively characterize the phytochemical composition of M. neglectum bulbs and evaluate their antioxidant and antiradical activity.

            The authors of manuscript ID: molecules-3923436 performed a phytochemical analysis of the composition of metabolites of the extract of the Mediterranean plant Muscari neglectum (Asparagaceae) Bulbs using the HPLC-ESI-qTOF-MS technique. Most of the compounds were annotated based on 72 precise masses, molecular formulas, their MS/MS fragmentation data, and comparison with literature databases of these compounds. However, a significant portion of the peaks shown in the chromatogram profile (Supplementary Material) remained unidentified. Among the 72 identified substances, various classes of natural metabolites were detected, including flavonoids, hydroxycinnamic acids, terpenoids, fatty acids, and triterpenoid saponins. Among these metabolites, flavonoids constituted the most abundant group, which included homoisoflavanones, which are characteristic metabolites of the genus Muscari, consistent with literature data. The authors determined total phenolic content, total flavonoid content and results are reported as milligrams of gallic acid equivalents per gram of dry extract using Folin–Ciocalteu assay in extracts obtained from M. neglectum bulbs.

            The authors determined the antioxidant properties of M. neglectum bulb extract by measuring its ferric reducing power (FRAP) and ABTS cation radical quenching capacity in trolox equivalent (TEAC), xanthine oxidase enzyme activity, and also assessed the free radical and reactive oxygen species (ROS) scavenging potential.

Comments

The authors did not study any biological activities, other than the antioxidant and antiradical properties of M. neglectum bulb extract, which possesses a broad spectrum of pharmacological properties, such as anti-inflammatory activity.

It is completely unclear which compounds (or classes of compounds) determine the pharmacological properties of M. neglectum bulb.

It is crucial that the work would be of interest to readers and useful for phytopharmaceuticals.

To achieve this goal, targeted fractionation (chromatography) of M. neglectum bulb extract with controlled bioactivity of the fractions is necessary.

The antioxidant, anti-inflammatory, and gastroprotective effects of these fractions should be studied. (Very often, the antioxidant and anti-inflammatory properties of flavonoids overlap.)

The active compounds (flavonoids or saponins) from the biologically active fractions should be isolated and identified.

Comments on the Quality of English Language

Review of the article

(Manuscript ID: molecules-3923436) “Phytochemical Characterisation and Antioxidant Potential of Muscari neglectum (Asparagaceae) Bulbs” by authors: María del Carmen Villegas-Aguilar, Antonio Segura-Carretero*, Víctor N. Suárez-Santiago

            Plants of the genus Muscari are known to have been used in Mediterranean and Balkan traditional medicine as diuretics, anti-inflammatory agents, and digestive aids, as well as for the treatment of kidney diseases. Their decoctions alleviate gastrointestinal discomfort and promote detoxification. The bulbs of Muscari comosum (L.) Mill were used as a remedy for toothache, and as anti-inflammatory, diuretic, and stimulant agents.

            Bulbs, as storage organs, are known to concentrate various classes of secondary metabolites, including flavonoids and saponins, which may be absent from aboveground tissues. Therefore, the authors state that the goal of their study is to comprehensively characterize the phytochemical composition of M. neglectum bulbs and evaluate their antioxidant and antiradical activity.

            The authors of manuscript ID: molecules-3923436 performed a phytochemical analysis of the composition of metabolites of the extract of the Mediterranean plant Muscari neglectum (Asparagaceae) Bulbs using the HPLC-ESI-qTOF-MS technique. Most of the compounds were annotated based on 72 precise masses, molecular formulas, their MS/MS fragmentation data, and comparison with literature databases of these compounds. However, a significant portion of the peaks shown in the chromatogram profile (Supplementary Material) remained unidentified. Among the 72 identified substances, various classes of natural metabolites were detected, including flavonoids, hydroxycinnamic acids, terpenoids, fatty acids, and triterpenoid saponins. Among these metabolites, flavonoids constituted the most abundant group, which included homoisoflavanones, which are characteristic metabolites of the genus Muscari, consistent with literature data. The authors determined total phenolic content, total flavonoid content and results are reported as milligrams of gallic acid equivalents per gram of dry extract using Folin–Ciocalteu assay in extracts obtained from M. neglectum bulbs.

            The authors determined the antioxidant properties of M. neglectum bulb extract by measuring its ferric reducing power (FRAP) and ABTS cation radical quenching capacity in trolox equivalent (TEAC), xanthine oxidase enzyme activity, and also assessed the free radical and reactive oxygen species (ROS) scavenging potential.

Comments

The authors did not study any biological activities, other than the antioxidant and antiradical properties of M. neglectum bulb extract, which possesses a broad spectrum of pharmacological properties, such as anti-inflammatory activity.

It is completely unclear which compounds (or classes of compounds) determine the pharmacological properties of M. neglectum bulb.

It is crucial that the work would be of interest to readers and useful for phytopharmaceuticals.

To achieve this goal, targeted fractionation (chromatography) of M. neglectum bulb extract with controlled bioactivity of the fractions is necessary.

The antioxidant, anti-inflammatory, and gastroprotective effects of these fractions should be studied. (Very often, the antioxidant and anti-inflammatory properties of flavonoids overlap.)

The active compounds (flavonoids or saponins) from the biologically active fractions should be isolated and identified.

Author Response

The authors did not study any biological activities, other than the antioxidant and antiradical properties of M. neglectum bulb extract, which possesses a broad spectrum of pharmacological properties, such as anti-inflammatory activity.

We thank the reviewer for this valuable comment. In the revised version of the manuscript, new experiments were conducted to evaluate the anti-inflammatory potential of the M. neglectum bulb extract. Specifically, two complementary assays were performed:

  1. Nitric oxide (·NO) radical scavenging assay, and
  2. Lipoxygenase (LOX) inhibition assay, using arachidonic acid as the substrate.

These analyses demonstrated that the extract effectively scavenged nitric oxide radicals (IC₅₀ = 78 ± 3 mg/L) and inhibited LOX activity (IC₅₀ = 66 ± 2 mg/L), supporting its anti-inflammatory potential. The experimental details have been added to the Materials and Methods section (Section 4.4.2 and 4.4.5), and the corresponding results are presented in Tables 6 and 7 (Section 2.4).

Additionally, these findings are now discussed in detail in the Discussion section (Lines 295–312), where we highlight that phenolic compounds and triterpenoid saponins—identified in the HPLC-ESI-qTOF-MS analysis—may contribute to the modulation of oxidative and enzymatic inflammatory pathways.

The Abstract and Conclusion have also been updated to reflect these new results.

We believe that the inclusion of these assays significantly strengthens the biological relevance of the study and provides a more comprehensive understanding of the pharmacological potential of M. neglectum bulbs

It is completely unclear which compounds (or classes of compounds) determine the pharmacological properties of M. neglectum bulb.

In the revised Discussion, we have clarified the relationship between chemical composition and pharmacological effects. Specifically, phenolic acids and flavonoids are primarily responsible for the antioxidant and radical scavenging activities of M. neglectum bulbs, while triterpenoid saponins contribute to anti-inflammatory effects by modulating eicosanoid biosynthesis and inhibiting LOX-mediated pathways. This information is now explicitly stated in the Discussion (Lines 313-320).

“Collectively, the pharmacological properties ….. medicinal uses of M. neglectum.”

It is crucial that the work would be of interest to readers and useful for phytopharmaceuticals. To achieve this goal, targeted fractionation (chromatography) of M. neglectum bulb extract with controlled bioactivity of the fractions is necessary.

In our study, we intentionally focused on the untargeted analysis of the whole M. neglectum bulb extract to capture the synergistic interactions among its compounds, which are often critical for the observed pharmacological effects. While targeted fractionation could provide information on individual components, our approach prioritizes the activity of the complete extract, which is directly relevant for phytopharmaceutical applications. We acknowledge that targeted fractionation may be a valuable future direction, and we have highlighted this in the revised Discussion as a potential avenue for further research. Thus, in the Discussion section, we have added the following paragraph to clarify this:

“In this study, we intentionally focused …. we note this as a potential avenue for further research.” (Lines 321-331)

The antioxidant, anti-inflammatory, and gastroprotective effects of these fractions should be studied. (Very often, the antioxidant and anti-inflammatory properties of flavonoids overlap.) The active compounds (flavonoids or saponins) from the biologically active fractions should be isolated and identified.

We agree that targeted fractionation and isolation of individual bioactive compounds could provide further insights into the specific contributions of flavonoids and saponins to antioxidant and anti-inflammatory effects. However, the present study intentionally focused on the complete M. neglectum bulb extract to evaluate potential synergistic interactions among its constituents, which are often critical for the overall pharmacological activity. This untargeted approach reflects a more holistic perspective that is directly relevant for phytopharmaceutical applications.

We acknowledge that fractionation and isolation of active compounds represent valuable future directions, and we have highlighted this in the revised Discussion as potential avenues for further research (Lines 321–331).

Reviewer 3 Report

Comments and Suggestions for Authors This manuscript focuses on the phytochemical analysis of aqueous-ethanol extracts of Muscari neglectum bulbs and the evaluation of their antioxidant activity. The authors used HPLC-MS/MS to identify 72 compounds and a comprehensive approach to assessing antioxidant properties using various biochemical assays. The use of standard antioxidants as controls increases the reliability of the results. Overall, this study contributes to the field of phytochemistry and antioxidant research. However, to improve the quality and reproducibility of the study, several significant caveats need to be addressed: 1) Limitations of the extraction method. Using only one solvent system (ethanol-water) significantly reduces the completeness and reproducibility of the phytochemical profile. It is recommended that the study be supplemented with extraction using solvents of varying polarity or that this limitation be discussed in detail in the text. 2) Lack of comparative analysis of plant organs. Conclusions regarding the specificity of the chemical composition of the bulbs are based on comparisons with published data on other plant organs grown under different conditions. For a reliable comparison, it is necessary to analyze the profiles of other organs (leaves, flowers) of plants from the same population. 3) The season of raw material collection is not specified. The "Materials and Methods" section lacks information on the time (season, month) of plant material collection. Since phytochemical composition can vary significantly depending on the phenological phase, this omission reduces the reproducibility of the study. This parameter must be specified. 4) Technical and editorial errors: a) An error in table numbering was found in the manuscript (there are two tables numbered 2). b) The text requires careful proofreading to eliminate typos (e.g., line 326). The manuscript is of scientific interest but requires revision.  

Author Response

This manuscript focuses on the phytochemical analysis of aqueous-ethanol extracts of Muscari neglectum bulbs and the evaluation of their antioxidant activity. The authors used HPLC-MS/MS to identify 72 compounds and a comprehensive approach to assessing antioxidant properties using various biochemical assays. The use of standard antioxidants as controls increases the reliability of the results. Overall, this study contributes to the field of phytochemistry and antioxidant research. However, to improve the quality and reproducibility of the study, several significant caveats need to be addressed: 

1) Limitations of the extraction method. Using only one solvent system (ethanol-water) significantly reduces the completeness and reproducibility of the phytochemical profile. It is recommended that the study be supplemented with extraction using solvents of varying polarity or that this limitation be discussed in detail in the text. 

We thank the reviewer for this observation. In our study, we intentionally focused on ethanol–water mixtures for extraction, as these solvents are considered safe, food-grade, and directly relevant for nutraceutical applications. Organic or highly nonpolar solvents, while potentially increasing the recovery of certain metabolites, are generally unsuitable for dietary supplements due to safety, regulatory, and formulation concerns.

To justify our choice, preliminary extractions were performed with several ethanol–water ratios (100:0, 80:20, 50:50, 20:80, 0:100, v/v) and the total phenolic content (TPC, mg GAE/g DE) was measured by the Folin–Ciocalteu assay. Absolute ethanol (100:0) yielded the highest TPC (96.3 ± 0.8 mg GAE/g DE). However, the 80:20 ethanol–water mixture was selected for subsequent analyses because:

  1. It preserves a substantial fraction of total phenolics (≈70% of the 100:0 value) while favouring the co-extraction of more polar phenolic species (e.g., glycosides) that are often of greater biological relevance.
  2. Hydroalcoholic extracts typically exhibit fewer non-phenolic reducing interferences in the Folin assay, improving the reliability of comparisons between samples.
  3. The 80:20 extract is more readily soluble in aqueous assay media and compatible with routine HPLC sample preparation, facilitating reproducible quantification and bioactivity testing.
  4. Practical considerations (safety, cost, and ease of solvent removal) further support the routine use of 80:20.

Taken together, these points justify the selection of the 80:20 ethanol–water mixture as the best compromise between extraction yield, selectivity for biologically relevant compounds, and downstream applicability for nutraceutical development.

In accordance with this, the manuscript has been updated to include Section 2.1, Preliminary Extraction Optimization (Lines 82–100) in the Results, and the extraction subsection of Materials and Methods has also been revised to describe these preliminary experiments and the rationale for selecting the 80:20 ethanol–water mixture for all subsequent analyses.

2) Lack of comparative analysis of plant organs. Conclusions regarding the specificity of the chemical composition of the bulbs are based on comparisons with published data on other plant organs grown under different conditions. For a reliable comparison, it is necessary to analyze the profiles of other organs (leaves, flowers) of plants from the same population. 

In response to the reviewer’s comment, the bulbs analyzed in this study were collected at an early developmental stage, when the plants had not yet produced leaves or flowers (as shown in Figure S6). Consequently, comparative analyses with other organs from the same individuals were not feasible.

Importantly, our focus on bulbs is justified because they act as storage organs, accumulating distinct secondary metabolites that are often not present or are present at lower levels in aerial tissues. Moreover, to the best of our knowledge, this is the first comprehensive phytochemical characterization of M. neglectum bulbs. While previous studies have analyzed leaves, flowers, or whole plants, specific investigations of bulbs are lacking. Our work thus provides novel insights into the chemistry of this species and highlights the pharmacological potential of its storage organs.

To clarify this point, additional information describing the developmental stage of the bulbs at the time of collection has been added to the Materials and Methods section (Lines 375–376), specifying that the samples were collected in October 2024, at the onset of germination.

3) The season of raw material collection is not specified. The "Materials and Methods" section lacks information on the time (season, month) of plant material collection. Since phytochemical composition can vary significantly depending on the phenological phase, this omission reduces the reproducibility of the study. This parameter must be specified.

The bulbs were collected in October 2024, at an early developmental stage when they were beginning to germinate (as illustrated in Figure S6). This information has been added to the Materials and Methods section (Lines 375-376) to ensure reproducibility and to provide context for the phytochemical composition of the samples.

4) Technical and editorial errors: a) An error in table numbering was found in the manuscript (there are two tables numbered 2). b) The text requires careful proofreading to eliminate typos (e.g., line 326). The manuscript is of scientific interest but requires revision.  

We thank the reviewer for noting these issues. The manuscript has been carefully revised throughout to correct typographical errors and inconsistencies. The table numbering error has been fixed, and the entire text has been thoroughly proofread to ensure accuracy, consistency, and clarity.

Round 2

Reviewer 1 Report

Comments and Suggestions for Authors

Dear Editor,

I have carefully reviewed the revised version of the manuscript. The authors have satisfactorily addressed all the comments and suggestions raised in the previous review round.  In my opinion, the manuscript is now suitable for publication in its present form.

Best regards,

Reviewer

Reviewer 2 Report

Comments and Suggestions for Authors

            In the revised version of the manuscript, "Phytochemical Characterization and Antioxidant Potential of Muscari neglectum (Asparagaceae) Bulbs (molecules-3923436)," the authors addressed all questions and comments. They conducted new experiments to evaluate the anti-inflammatory potential of M. neglectum bulb extract. For this purpose, two additional assays were conducted: a nitric oxide (NO) radical scavenging assay and a lipoxygenase (LOX) inhibition assay using arachidonic acid as a substrate. The new data obtained significantly strengthened the biological significance of the study and provided a more complete understanding of the pharmacological potential of M. neglectum bulbs.

            In the "Discussion" section, the authors further discussed the relationship between the chemical composition and pharmacological effects of the identified natural compounds and indicated which classes of substances may mediate the anti-inflammatory effect of M. neglectum bulb extract. The revised version of the manuscript "Phytochemical Characterisation and Antioxidant Potential of Muscari neglectum (Asparagaceae) Bulbs (molecules-3923436)" is significantly more suitable for publication in Molecules.

Comments on the Quality of English Language

Review of the article

(Manuscript ID: molecules-3923436) “Phytochemical Characterisation and Antioxidant Potential of Muscari neglectum (Asparagaceae) Bulbs” by authors: María del Carmen Villegas-Aguilar, Antonio Segura-Carretero*, Víctor N. Suárez-Santiago

            Plants of the genus Muscari are known to have been used in Mediterranean and Balkan traditional medicine as diuretics, anti-inflammatory agents, and digestive aids, as well as for the treatment of kidney diseases. Their decoctions alleviate gastrointestinal discomfort and promote detoxification. The bulbs of Muscari comosum (L.) Mill were used as a remedy for toothache, and as anti-inflammatory, diuretic, and stimulant agents.

            Bulbs, as storage organs, are known to concentrate various classes of secondary metabolites, including flavonoids and saponins, which may be absent from aboveground tissues. Therefore, the authors state that the goal of their study is to comprehensively characterize the phytochemical composition of M. neglectum bulbs and evaluate their antioxidant and antiradical activity.

            The authors of manuscript ID: molecules-3923436 performed a phytochemical analysis of the composition of metabolites of the extract of the Mediterranean plant Muscari neglectum (Asparagaceae) Bulbs using the HPLC-ESI-qTOF-MS technique. Most of the compounds were annotated based on 72 precise masses, molecular formulas, their MS/MS fragmentation data, and comparison with literature databases of these compounds. However, a significant portion of the peaks shown in the chromatogram profile (Supplementary Material) remained unidentified. Among the 72 identified substances, various classes of natural metabolites were detected, including flavonoids, hydroxycinnamic acids, terpenoids, fatty acids, and triterpenoid saponins. Among these metabolites, flavonoids constituted the most abundant group, which included homoisoflavanones, which are characteristic metabolites of the genus Muscari, consistent with literature data. The authors determined total phenolic content, total flavonoid content and results are reported as milligrams of gallic acid equivalents per gram of dry extract using Folin–Ciocalteu assay in extracts obtained from M. neglectum bulbs.

            The authors determined the antioxidant properties of M. neglectum bulb extract by measuring its ferric reducing power (FRAP) and ABTS cation radical quenching capacity in trolox equivalent (TEAC), xanthine oxidase enzyme activity, and also assessed the free radical and reactive oxygen species (ROS) scavenging potential.

Comments

The authors did not study any biological activities, other than the antioxidant and antiradical properties of M. neglectum bulb extract, which possesses a broad spectrum of pharmacological properties, such as anti-inflammatory activity.

It is completely unclear which compounds (or classes of compounds) determine the pharmacological properties of M. neglectum bulb.

It is crucial that the work would be of interest to readers and useful for phytopharmaceuticals.

To achieve this goal, targeted fractionation (chromatography) of M. neglectum bulb extract with controlled bioactivity of the fractions is necessary.

The antioxidant, anti-inflammatory, and gastroprotective effects of these fractions should be studied. (Very often, the antioxidant and anti-inflammatory properties of flavonoids overlap.)

The active compounds (flavonoids or saponins) from the biologically active fractions should be isolated and identified.

Reviewer 3 Report

Comments and Suggestions for Authors In this form the manuscript is suitable for publication.